# EDM2+: Exploring Efficient Diffusion Model Architectures for Visual Generation

## Abstract

The training and sampling of diffusion models have been exhaustively elucidated in prior art (Karras et al., 2022; 2024b). Instead, the underlying network architecture design remains on a shaky empirical footing. Furthermore, in accordance with the recent trend of scaling law, large-scale models make inroads into generative vision tasks. However, running such large diffusion models incurs a sizeable computational burden, rendering it desiderata to optimize calculations and efficiently allocate resources. To bridge these gaps, we navigate the design landscape of efficient U-Net based diffusion models, stemming from the prestigious EDM2. Our exploration route is organized along two key axes, layer placement and module interconnection. We systematically study fundamental design choices and uncover several intriguing insights for superior efficacy and efficiency. These findings culminate in our redesigned architecture, EDM2+, that reduces the computational complexity of the baseline EDM2 by $2\times$ without compromising the generation quality. Extensive experiments and comparative analyses highlight the effectiveness of our proposed network architecture, which achieves the state-of-the-art FID on the hallmark ImageNet benchmark. Code will be released upon acceptance.

## 1 Introduction

In recent years, diffusion models have swept the field of generative modeling, catalyzing a plethora of applications to image (Rombach et al., 2022; Podell et al., 2024; Esser et al., 2024), video (Ho et al., 2022; Blattmann et al., 2023), and 3D shape generation (Poole et al., 2023; Wang et al., 2023) in the realm of visual synthesis. Dating back to a decade ago, the advent of diffusion models relies on a plain Convolutional Neural Network (CNN) architecture (Sohl-Dickstein et al., 2015). The embarrassingly simple architecture might have posed a hindrance to the immediate blossoming of diffusion models. During the following period, one has witnessed a meteoric rise of Generative Adversarial Networks (GAN) (Goodfellow et al., 2014) in yielding photorealistic imagery (Karras et al., 2018; 2019; 2020b; 2021). In the meantime, the development of diffusion models is sluggish but has never stood still. Until 2020s, Denoising Diffusion Probabilistic Model (DDPM) (Ho et al., 2020) resurges, sparking a new wave of deep generative modeling. In this seminal work, DDPM, the introduction of U-Net (Ronneberger et al., 2015) backbone in tandem with a few modern architectural components (*e.g.*, Group Normalization (Wu & He, 2018), self-attention, and position embedding (Vaswani et al., 2017)) unleashes the potential of diffusion models in producing high-quality images comparable to other types of generative models. Henceforth, diffusion models make tremendous strides forward within the ambit of visual generative modeling.

Note that the initial adoption of U-Net in diffusion modeling borrows from other established templates, *i.e.*, its successful practice in Pixel-CNN++ (Salimans et al., 2017). Coincidentally, Jolicoeur-Martineau et al. (2021) reveal that U-Net performs substantially better than RefineNet (Lin et al., 2017) which is extensively utilized by score-based generative models (Song & Ermon, 2019; 2020). These independent observations disclose the pivotal role of network architecture design in facilitating generative modeling. In light of that, follow-up works, including iDDPM (Nichol & Dhariwal, 2021) and ADM (Dhariwal & Nichol, 2021), continuously polish the network architecture, thereby elevating the performance upper bound. These aforementioned architectures are primarily grounded on convolution operators. More recently, pure attention-based architecture further enriches the network design space, represented by Diffusion Transformer (DiT) (Peebles & Xie, 2023). The groundbreaking Sora (Brooks et al., 2024) also adopts a spatiotemporal DiT as the foundation model for

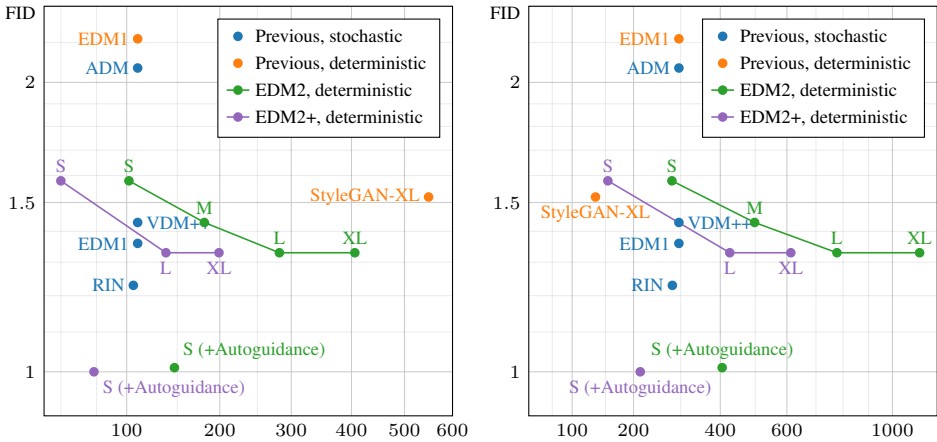

Figure 1: Our architecture EDM2+ achieves performance parity with EDM2 using $2\times$ less compute across a wide spectrum of model sizes without guidance. Armed with the latest Autoguidance, our model is located at the bottommost leftmost corner among an array of generative architectures. In this plot, we use gigaflops per single model evaluation as a criterion of a model's intrinsic computational complexity, a similar advantage keeps consistent in terms of parameter count.

text-to-video generation. Simultaneously, Stable Diffusion 3 (Esser et al., 2024) employs a multi-modal DiT (MMDiT) as the base architecture for text-to-image generation.

On the one hand, notwithstanding the flexibility and scalability of Diffusion Transformer, the final generation quality is largely dictated by its voracious appetite for the training resource. On the other hand, the top-performing diffusion models built upon relatively lightweight CNNs still prevail, *e.g.*, EDM2 (Karras et al., 2024b) showcases performance lead over Transformer-based architectures on ImageNet. Overall, EDM2 copies the U-Net macro architecture acknowledged by preceding works. Despite a few retouches of the network layers, the existing micro architecture is yet underexplored and there leaves considerable room for advanced architecture design accordingly. To fulfill this gap, we delve into the design principles and architectural choices critical for visual generation via in-depth analysis and ablation experiments. Benefiting from its streamlined and modular architecture, EDM2, as a good starting point, would ease the exploration of network design space. Startng from EDM2, we launch our investigation from the perspective of layer arrangement and inter-module connection, and eventually craft a tailored model architecture, coined as EDM2+, with on-par or even better generation quality and enhanced efficiency compared with the EDM2 counterpart.

In our design roadmap (§3), we apply the changes step-by-step to the EDM2 architecture and evaluate the impact of these individual ingredients. Our findings are in general two-fold: first, decomposing the spatial/channel mixing operations and shifting the computation focus from spatial to channel dimension strikes a better balance; second, through the lens of information bottleneck, contracting the output dimension of the embedding network concentrates the most expressive condition information to facilitate the entire information flow and naturally diminishes the parameter amount. The above conclusions are then materialized in an innovative network block, which encompasses a sequence of depthwise and pointwise convolutions, with the condition embedding sandwiched between the narrow convolution layers, as visualized in Figure 3. Our EDM2+ model architecture is comprised of dozens of such building blocks, outperforming existing top-tier diffusion models and GANs in the FID evaluation metric while remarkably reducing the model computation and storage consumption, as portrayed in Figure 1. Equipped with better utilization of guidance (Karras et al., 2024a), our endeavor sets new record FID on the ImageNet $64 \times 64$ benchmark, albeit using fast deterministic sampling.

Our core contributions could be summarized as:

- We conduct comprehensive experiments on the basis of EDM2 and meticulously identify the limitations in the crucial architecture components.

- We further conceptualize performance-optimized solutions, for the purpose of strengthening both the generation quality and efficiency.

- The devised architecture EDM2+ excels other leading diffusion models and GANs on the ImageNet benchmark, offering a new standard to the generative modeling field.

## 2    RELATED WORK

We provide a skim-through of several important aspects revolving around diffusion models in prior literature, spanning from training and sampling to network architecture design. We also clarify their similarities and differences compared with our work.

### 2.1    DIFFUSION TRAINING

Drawing inspiration from nonequilibrium thermodynamics (Sohl-Dickstein et al., 2015), diffusion models decompose the entire generative process into progressive denoising transitions from standard Gaussian noise to clean images. In stark contrast to other explicit likelihood-based models (*e.g.*, Variational AutoEncoder (Kingma & Welling, 2014), Autoregressive models (van den Oord et al., 2016; Salimans et al., 2017), and Non-Autoregressive models (Chang et al., 2022; Yu et al., 2023)) or implicit likelihood-based models (*e.g.*, GAN (Goodfellow et al., 2014)), diffusion models pose the generation task as a supervised learning scheme, greatly enhancing training stability and thus scalability. In practice, the likelihood-induced Evidence Lower Bound (ELBO) is simplified to an $\ell_2$ regression learning objective. Depending on this realization, different regression targets, including image (Sohl-Dickstein et al., 2015), noise (Ho et al., 2020), and velocity (Salimans & Ho, 2022), simply translate to different loss function weights (Kingma & Gao, 2023). In consequence, scaling up diffusion models to billions of parameters and web-scale training data becomes more frictionless compared to the previous prevalent GANs (Kang et al., 2023), incubating a bunch of text-to-image commercial products, such as Stable Diffusion series (Rombach et al., 2022; Podell et al., 2024; Esser et al., 2024), DALL·E 2&3 (Nichol et al., 2022; Ramesh et al., 2022; Betker et al., 2023), and Imagen series (Saharia et al., 2022; Imagen-Team-Google et al., 2024). In the present work, we inherit the EDM (Karras et al., 2022) preconditioning framework for training due to its well-behaved training dynamics.

### 2.2    DIFFUSION SAMPLING

For diffusion models, the sampling procedure typically demands thousands of consecutive steps to synthesize a high-quality image. Theoretically, the diffusion backward process could be interpreted as a reverse Stochastic Differential Equation (SDE) or the corresponding Probability Flow Ordinary Differential Equation (PF-ODE) (Song et al., 2021b). Therefore, there naturally exists a trade-off between the discretion error and step size. The straighter the sampling trajectory, the larger step size can be tolerated. As such, much endeavor has been devoted to straightening the trajectory (Song et al., 2021a; Karras et al., 2022; Liu et al., 2023) and inventing advanced ODE solvers (Liu et al., 2022a; Lu et al., 2022; Zhang & Chen, 2023). In parallel, a growing body of diffusion distillation techniques (Salimans & Ho, 2022; Meng et al., 2023; Luo et al., 2023; Yin et al., 2024) is proposed to reduce the number of sampling steps. In addition, the overall sampling cost could also be reduced by cutting down the model latency per step in the denoising trajectory. Our work goes along this research vein, contributing to a compact yet high-performing model via reworking the network structure from scratch. Hence, our work distinguishes itself from post-hoc pruning (Li et al., 2023b), quantization (Li et al., 2023a) and cache (Wimbauer et al., 2024) methodology but is complementary.

### 2.3    NETWORK ARCHITECTURE ENGINEERING

U-Net is ubiquitously applied to low-level vision tasks, including visual segmentation (Chen et al., 2018), generation (Kingma et al., 2016) and so on. Skip connections in the network always play an instrumental role in transmitting the high-resolution signal to the output end for detailed refinement, either in CNN or Transformer (Bao et al., 2023). Building upon a U-Net backbone, DDPM (Ho et al.,

2020) interleaves convolution blocks with self-attention modules (Vaswani et al., 2017), effectively gathering long-range pixel dependence. iDDPM (Nichol & Dhariwal, 2021) extends single-head self-attention to multi-head ones and widens its usage over a broader range of feature resolutions. Adaptive Group Norm (AdaGN) is involved as well, resembling AdaIN (Huang & Belongie, 2017). ADM (Dhariwal & Nichol, 2021) additionally steals the network topology and scaled residual connections from the GAN literature (Brock et al., 2019; Karras et al., 2020b). Several hyperparameters are ablated here, including the network depth/width and the number of attention heads. EDM2 (Karras et al., 2024b) emphasizes standardizing the magnitudes of network weights, activations, gradients, *etc.*, in the same spirit of pioneering magnitude-focusing image recognition networks (Brock et al., 2021a;b). Diffusion Transformers (DiT) position themselves as appealing alternatives to the *de facto* standard U-Net, attracting enthusiasm from both academia and industry (Peebles & Xie, 2023; Hoogeboom et al., 2023). Subsequently, DiffuSSM (Yan et al., 2024) supplants the attention mechanism of DiT with the State Space Model (SSM) (Gu et al., 2022) blocks to promote the efficiency. Our work takes root in a hybrid architecture, EDM2, that alternates convolution blocks with attention modules and manifests as an outstanding denoiser architecture.

## 3 DESIGN ROADMAP

We first give a brief recap on the network design of preeminent diffusion models. Next, we provide our intuition about model design and present the roadmap to an efficient architecture while preserving the generation quality, in which the design path could be split into two branches, layer placement in the denoising network and its synergy with the embedding network.

### 3.1 PRELIMINARY

EDM1 and EDM2 commonly employ the encoder-decoder paradigm in the denoising network, into which a noise-perturbed image is fed and from which a reparameterized denoised image is retrieved. Each network block stacks two $3 \times 3$ regular convolutions for feature extraction or mixing. Differently, Group Normalization (Wu & He, 2018) and SiLU nonlinearity (Ramachandran et al., 2018) precede convolution in EDM1 while EDM2 revokes such normalizations and substitutes SiLU with MP-SiLU. A skip connection is indispensable to facilitate smooth training (He et al., 2016). Formally, a network block either in the encoder or decoder of EDM2 could be formulated as

$$\boldsymbol{o} = \mathrm{conv}(\varphi(\mathrm{conv}(\varphi(\boldsymbol{x})))) + \gamma(\boldsymbol{x}), \tag{1}$$

where $\varphi$ denotes the MP-SiLU activation function and $\gamma$ refers to a potential linear projection that compresses channels only in the decoder part (otherwise it is identity in the encoder part). $\boldsymbol{o}$ and $\boldsymbol{x}$ enumerate the input and output feature tensor of the considered network block. Moreover, for conditional generation, a condition embedding is utilized to rectify the midway representation, written as

$$\boldsymbol{o} = \mathrm{conv}(\varphi(\mathrm{conv}(\varphi(\boldsymbol{x}) \times \phi(\boldsymbol{c})))) + \gamma(\boldsymbol{x}), \tag{2}$$

where $\boldsymbol{c}$ defines the condition information and $\phi$ symbolizes the mapping network that transforms it into a high-dimensional embedding. A self-attention module is suggested to append to this convolution block in case that the feature resolution is low, for example, $^{1}/_{4}$ and $^{1}/_{8}$ of the input noisy image resolution. Since self-attention does not occupy a majority of the computation in our context, we omit to discuss it in the sequel.

Reminiscent of the classical Progressive GAN (Karras et al., 2018), EDM2 draws a wealth of lessons from it: standard normal weight initialization, constant input channel concatenation, pixel normalization, and magnitude-preserving learned layers (aka equalized learning rate). EDM2 also echoes certain modern architecture design philosophies: pixel norm analogous to RMSNorm (Zhang & Sennrich, 2019), magnitude-preserving layers to weight standardization (Qiao et al., 2020) and cosine attention to QK Normalization (Dehghani et al., 2023). In turn, EDM2 now serves as a starting point for our redesigned architecture. To hit our ultimate design, we also additionally tap wisdom from other celebrated model architectures.

We shall revise the existing network details and present thorough experimental results of our exploration journey in Table 1 and 2. Concretely, the revisions are enabled one by one on top of the baseline EDM2 and we delineate the stepwise quality metric accompanied by the model parameters and computational complexity. Figure 2 and 3 sketch the architecture layout of each intermediate

Table 1: Ablated architectures of the denoising network block. $^{\dagger}$ reproduction with the official code. "c" stands for the base channel number of the first block in the entire network, while "e" the channel expansion ratio of the first pointwise convolution inside a block and "k" the kernel size of the only depthwise convolution. The same in Table 2.

| Architecture | Mparams | GFLOPs | FID-50K |
|---|---|---|---|
| baseline (original publication) | 280.21 | 101.90 | 1.58 |
| A conv, c192 (baseline reproduced$^{\dagger}$) | 280.21 | 101.90 | 1.63 |
| B dwconv, c384 | 252.28 | 51.13 | 1.81 |
| C dsconv, e6 | 273.99 | 72.52 | 1.75 |
| D dsconv, e6, linear bottleneck | 273.99 | 72.44 | 1.57 |
| D$^{\star}$ dsconv, e4, linear bottleneck | 195.59 | 51.09 | 1.60 |
| E mbconv, e6 | 273.76 | 72.27 | 1.63 |
| E$^{\diamond}$ mbconv, e6, k7 | 278.05 | 75.27 | 1.64 |
| F +dwconv at the end | 195.76 | 51.26 | 1.68 |
| G −dwconv in the middle | 195.11 | 50.61 | 1.66 |

configuration. We perform our evaluation on the class-conditional ImageNet (Deng et al., 2009) $64 \times 64$ dataset, with the identical training recipe and data processing strategy to EDM2, in order to isolate the influence of network design. For fast prototyping, we choose a modest-sized model, EDM2-S with approximately 300M trainable parameters as the baseline (reproduced by ourselves as **CONFIG A** in Table 1), with more results for scaled-up models presented later. We follow the evaluation protocol of common practice and measure the final performance with Fréchet Inception Distance (FID) (Heusel et al., 2017) on 50,000 synthesized images (*i.e.*, FID-50K). We defer more implementation details to Section 4.

## 3.2 DENOISING NETWORK DESIGN

Our overarching goal is to slim the model architecture without prejudice to the generation quality. Recall that previous lightweight network designs usually resort to a couple of depthwise and pointwise convolutions, supporting spatial and channel information mixing respectively. On this premise, we further posit that at the heart of a visual synthesis task is not only spatial pattern mixing or refinement but also composing semantically meaningful components for a high-fidelity imagery. In essence, this task is complicated by reasoning from the interaction between scene and objects, which cannot be solely reflected from superficial spatial patterns. Thus, once within a tight computational budget, we advocate trading the spatial representation mixing for stronger semantic representation learning in the channel dimension.

First, as a pilot experiment, we replace the regular convolution with depthwise convolution. Meanwhile, the channel number throughout the entire network is doubled to keep a reasonable model capacity. Then, each building block can be derived as

$$\boldsymbol{o} = \texttt{dwconv}(\varphi(\texttt{dwconv}(\varphi(\boldsymbol{x})) \times \phi(\boldsymbol{c}))) + \gamma(\boldsymbol{x}). \qquad (3)$$

We observe that after roughly halving the computational complexity, the FID sacrifices not too much, which is shown as **CONFIG B** in Table 1. It indicates the relative importance of spatial and channel mixing, prompting us to allocate more computing resources to channel mixing.

Second, we speculate the optimal way to arrange the computation of channel mixing is not evenly distribute it to all layers as above. Following renowned models for efficient network design including Xception (Chollet, 2017), MobileNet series (Howard et al., 2017; Sandler et al., 2018; Howard et al., 2019; Qin et al., 2024) and EfficientNet series (Tan & Le, 2019; 2021), we substitute depthwise separable convolution (dsconv) for regular convolution, where we only expand the channel dimension in the first pointwise convolution (expansion ratio set to 6). The building block now becomes

$$\boldsymbol{o} = \texttt{pwconv}(\varphi(\texttt{dwconv}(\varphi(\texttt{pwconv}(\varphi(\texttt{dwconv}(\varphi(\boldsymbol{x}))))) \times \phi(\boldsymbol{c}))) + \gamma(\boldsymbol{x}). \qquad (4)$$

This variant achieves a decent performance but is still not satisfactory, demonstrated as **CONFIG C** in Table 1. Motivated by the linear bottleneck principle in MobileNetV2 (Sandler et al., 2018), we

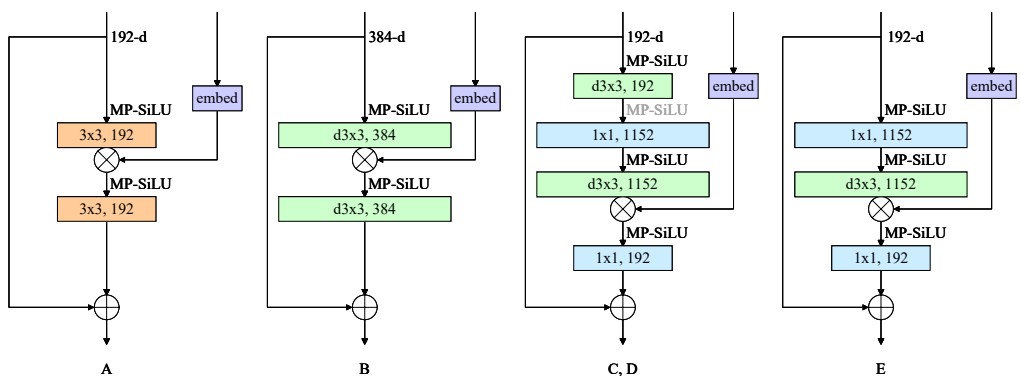

Figure 2: Block specifications of CONFIGS A–D. A (EDM2 baseline) regular convolution. B depthwise convolution. C, D depthwise separable (depthwise + pointwise) convolution, MP-SiLU in gray indicates only existence in CONFIG C. E mobile convolution as MobileNetV2 (Sandler et al., 2018). The width of each layer is proportional to the number of channels. Best viewed in color.

remove the MP-SiLU in the narrow layers[1]. It also accords with the argument of "fewer activation functions" in ConvNeXt (Liu et al., 2022b). This simple modification results in the following transformation

$$\boldsymbol{o} = \mathtt{pwconv}(\varphi(\mathtt{dwconv}(\varphi(\mathtt{pwconv}(\mathtt{dwconv}(\varphi(\boldsymbol{x}))))) \times \phi(\boldsymbol{c}))) + \gamma(\boldsymbol{x}), \quad (5)$$

and effectively mitigates the representation bottleneck. Hence, **CONFIG D** in Table 1 improves the quality metric obviously compared to CONFIG C and has already surpassed the baseline CONFIG A.

### 3.3 QUO VADIS, SPATIAL MIXING PRIMITIVES?

All the above taken into account, a question arises: to what extent do spatial mixing primitives work? To answer this question, we fade in or out depthwise convolution to scrutinize its impact.

*Fade Out* We erase one depthwise convolution at the start of CONFIG C block, leaving only a single depthwise convolution and leading to a lite edition CONFIG E,

$$\boldsymbol{o} = \mathtt{pwconv}(\varphi(\mathtt{dwconv}(\varphi(\mathtt{pwconv}(\varphi(\boldsymbol{x}))))\times \phi(\boldsymbol{c}))) + \gamma(\boldsymbol{x}). \quad (6)$$

We find that the generation quality holds, corroborating our hypothesis that channel mixing operations outweigh spatial ones under a limited computational budget. Intriguingly, we notice that at this moment the network block looks pretty like MBConv (Tan et al., 2019), so we abuse the notation to term **CONFIG E** as mbconv in Table 1.

In addition, there appears a tendency to employ larger kernel sizes, such as $5 \times 5$ (Tan et al., 2019) or $7 \times 7$ (Liu et al., 2022b). We make an attempt to enlarge the kernel size of the only depthwise convolution to $7 \times 7$ but observe a neutral effect in terms of the quality metric, shown as **CONFIG E$^\diamond$** in Table 1. This phenomenon is possibly attributed to the existence of self-attention that has already effectively captured long-range feature correspondences.

As an episode, we step a little back to scale down the channel expansion ratio of the current best-performing variant CONFIG D from 6 to 4, so as to constrain the computational cost to nearly 50% of the baseline. This action makes the subsequent experiments more affordable and guarantees that the generation quality is still superior to the baseline, marking a promising checkpoint **CONFIG D$^\star$**. We shall engage with CONFIG D$^\star$ in the remaining.

*Fade In* We attach another depthwise convolution at the end of CONFIG D$^\star$ block, calculated as

$$\boldsymbol{o} = \mathtt{dwconv}(\varphi(\mathtt{pwconv}(\varphi(\mathtt{dwconv}(\varphi(\mathtt{pwconv}(\mathtt{dwconv}(\varphi(\boldsymbol{x}))))) \times \phi(\boldsymbol{c}))))) + \gamma(\boldsymbol{x}), \quad (7)$$

which even causes performance regression, illustrated as **CONFIG F** in Table 1.

---

[1] narrow means a small channel dimension while wide means a large one, following Sandler et al. (2018).

Table 2: Ablated architectures of the interplay between embedding and denoising networks.

| Architecture | Mparams | GFLOPs | FID-50K |
|---|---|---|---|
| A conv, c192 (baseline) | 280.21 | 101.90 | 1.63 |
| D⋆ dsconv, e4, linear bottleneck | 195.59 | 51.09 | 1.60 |
| F +dwconv at the end | 195.76 | 51.26 | 1.68 |
| F* +dwconv at the end, embed bottleneck | 154.61 | 51.05 | 1.60 |
| G −dwconv in the middle | 195.11 | 50.61 | 1.66 |
| G* −dwconv in the middle, embed bottleneck | 153.97 | 50.41 | 1.58 |

*Fade Out* Given the redundancy of depthwise convolution in CONFIG F, one of the three depthwise convolutions could be safely removed without observable drawbacks. Since eliminating the depthwise convolution at the starting position does not provide a clear gain (remember CONFIG E *vs.*CONFIG D), it is tentative to exclude the one in the middle, described as **CONFIG G**

$$\boldsymbol{o} = \text{dwconv}(\varphi(\text{pwconv}(\varphi(\text{pwconv}(\text{dwconv}(\varphi(\boldsymbol{x}))) \times \phi(\boldsymbol{c}))))) + \gamma(\boldsymbol{x}). \tag{8}$$

It gives rise to slightly improved performance and reduced computation overhead, as validated in Table 1.

In response to the question raised in the beginning: excessive spatial mixing primitives are indeed unnecessary for a better generation quality (CONFIG F), while too few of them deteriorate the performance (CONFIG E). Therefore, CONFIG G is taken for granted in the following exploration.

### 3.4 EMBEDDING NETWORK DESIGN

The condition embedding is the crux of injecting external condition signals into the main stream of the denoising network. The condition information might be a timestamp (Ho et al., 2020) or a noise level (Song & Ermon, 2019), a class label (Dhariwal & Nichol, 2021) or a more verbose textual description (Rombach et al., 2022). In this work, we operate on the noise level and the class label information, since our exploration is set out on the ImageNet benchmark for class-conditioned image synthesis. Typically, an input numeral, representative of the condition information, is appropriately scaled and mapped to a high-dimensional space using non-learnable Fourier feature (Tancik et al., 2020) or sinusoidal embedding, in concert with a (few) learnable linear projection layer(s). This stack of neural layers is collectively dubbed as the embedding network.

It is trendy that the embedding network is gradually minimized, exemplified by the shallower mapping network in StyleGAN2-ADA (Karras et al., 2020a) or StyleGAN3 (Karras et al., 2021) and the trimmed embedding network in EDM2 (Karras et al., 2024b). The model capacity of this tiny network is presumably sufficient to extract semantic information from a single scalar condition, while a shorter path here permits the denoising network to be better informed of the condition information. Provided the network depth is extremely truncated by design, we are particularly interested in how to maximize its cooperation with the denoising network from other factors, for instance, whether the network width of their junction makes a difference to the generation quality and parameter efficiency.

Regarding the above CONFIGS F–G, the condition embedding is mixed with a *wide* feature map. The conventional wisdom is that the condition embedding is responsible for steering the style of synthesized images in a global manner (Huang & Belongie, 2017; Karras et al., 2019). From the viewpoint of information bottleneck theory, integrating such condition information into a *narrow* feature map would be more effective, yielding more targeted and predictable control of the entire information flow. To substantiate our hypothesis, we reposition the embedding network after the last pointwise convolution, constructing an "embed bottleneck" in **CONFIG G*** (similar for **CONFIG F***)

$$\boldsymbol{o} = \text{dwconv}(\varphi(\text{pwconv}(\varphi(\text{pwconv}(\text{dwconv}(\varphi(\boldsymbol{x}))))) \times \phi(\boldsymbol{c}))) + \gamma(\boldsymbol{x}). \tag{9}$$

Notably, this step shoots two hawks with one arrow, not only reducing the network parameters by over 20% but also meliorating the generation quality. Thanks to the boosted bottleneck representation, the FID metric is again elevated beyond the baseline CONFIG A, as exhibited in Table 2 CONFIG F* and CONFIG G*.

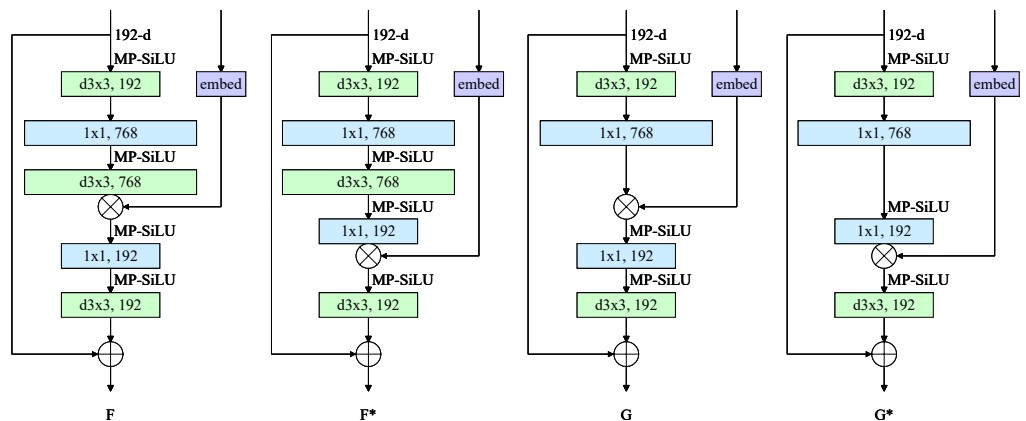

Figure 3: Block specifications of CONFIGS F–G. CONFIGS F*–G* rewire the embedding network to a narrow feature tensor in the denoising network, shaping an "embed bottleneck". G* ours EDM2+.

In a nutshell, the key to both efficacy and efficiency is modulating the condition embedding into the denoising network's bottleneck layers. A proof by contradiction occurs in CONFIG E, where there are two optional positions for embedding modulation, that are, following the first pointwise convolution or the first depthwise convolution. Indeed, we do not find a noticeable difference between them (with the same FID of 1.63). It is supposed that both feature maps are of equal width, without a bottleneck of information flow in the mainstream denoising network.

## 4 EXPERIMENTS

### 4.1 DATASETS

We adopt ImageNet (Deng et al., 2009) pixel-space diffusion at $64 \times 64$ resolution as the benchmark. Before training, we follow ADM (Dhariwal & Nichol, 2021) protocol to pre-process the raw ImageNet dataset for a fair comparison to previous works. Specifically, the images are resized along the short edge and then cropped at the center to a desired square shape. No data augmentation is applied during training, since a large-scale dataset like ImageNet is deemed to be challenging enough to fit for most visual generation models.

### 4.2 EVALUATION

The evaluated checkpoints are constructed post-hoc through power-function Exponential Moving Average (EMA) over a group of snapshots with the recommended length by EDM2. The evaluation metric is the widely recognized FID (Heusel et al., 2017). It compares the distribution statistics of 50K synthesized samples against all the 1,281,167 real images in the training dataset, in line with common practice. The class labels for the 50K synthesized images are drawn from a uniform distribution. For feature extraction, we use the pre-trained Inception-v3 (Szegedy et al., 2016) model provided by StyleGAN3 (Karras et al., 2021). Limited by the computational resource, we compute FID only once, which may even put us at a disadvantage in comparison to EDM2 (because EDM2 computes FID three times and reports the *minimum*).

### 4.3 IMPLEMENTATION DETAILS

We implemented our network architecture based on the PyTorch (Paszke et al., 2019) library and EDM2 codebase[2]. All training runs are conducted on 32 NVIDIA A100-SMX4-80G GPUs, while each evaluation run is executed on a single node with 8 GPUs. The entire training traverses either

---

[2] https://github.com/NVlabs/edm2

Table 3: State-of-the-art comparison on ImageNet at $64 \times 64$ resolution. NFE states the Number that the score Function is Evaluated to synthesize a single image. ↓ hints lower is better. GFLOPs tell the floating-point operations per function call. The entries with Autoguidance combine an S-sized model with an XS-sized unconditional one, taking the guidance model's cost into consideration.

| Architecture | Deterministic | | Stochastic | | Model size | |
|---|---|---|---|---|---|---|
| | FID↓ | NFE | FID↓ | NFE | Mparams | Gflops |
| ADM (Dhariwal & Nichol, 2021) | – | – | 2.07 | 250 | 296 | 110 |
| + EDM1 sampling (Karras et al., 2022) | 2.66 | 79 | 1.57 | 511 | 296 | 110 |
| + EDM1 training (Karras et al., 2022) | 2.22 | 79 | 1.36 | 511 | 296 | 110 |
| VDM++ (Kingma & Gao, 2023) | – | – | 1.43 | 511 | 296 | 110 |
| RIN (Jabri et al., 2023) | – | – | 1.23 | 1000 | 281 | 106 |
| StyleGAN-XL (Sauer et al., 2022) | 1.52 | 1 | – | – | 134 | 549 |
| EDM2-S (Karras et al., 2024b) | 1.58 | 63 | – | – | 280 | 102 |
| + Autoguidance(XS, $T/8$) (Karras et al., 2024a) | 1.01 | 63 | – | – | 405 | 147 |
| EDM2-M (Karras et al., 2024b) | 1.43 | 63 | – | – | 498 | 181 |
| EDM2-L (Karras et al., 2024b) | 1.33 | 63 | – | – | 777 | 282 |
| EDM2-XL (Karras et al., 2024b) | 1.33 | 63 | – | – | 1119 | 406 |
| EDM2+-S | 1.58 | 63 | – | – | 154 | 50 |
| + Autoguidance(XS, $T/8$) (Karras et al., 2024a) | 1.00 | 63 | – | – | 213 | 73 |
| EDM2+-L | 1.33 | 63 | – | – | 426 | 138 |
| EDM2+-XL | 1.33 | 63 | – | – | 613 | 199 |

2147.5M or 671.1M images with a mini-batch size of 64 per device. We adopt the Adam (Kingma & Ba, 2015) optimizer with a peak learning rate of ∼0.01 and constant betas $\beta_1 = 0.9, \beta_2 = 0.99$. The learning rate is linearly warmed up over the first 10M images and decayed after 70K training iterations following a reciprocal square root schedule (Zhai et al., 2022). Larger models enjoy a moderately lower learning rate and higher dropout rate. Mixed-precision training (Micikevicius et al., 2018) is allowed to take full advantage of the tensor cores in NVIDIA Ampere architecture. Almost all activation values are cast to the 16-bit floating point (FP16) format during network forward/backward. To avoid the risk of under/overflows, it is sufficient to only cast the `NaN` and `Inf` gradient values to zeros. The second-order Heun sampler is adopted for ODE sampling, with all the hyperparameters aligned with the original EDM1 setup. The EDM2+-S model is built upon network blocks of **CONFIG G\*** in Figure 3. The L-sized and XL-sized versions are obtained by scaling up the network width of EDM2+-S to 320 and 384 respectively.

### 4.4 QUANTITATIVE RESULTS

*Comparison to deterministic sampling.* As depicted in Table 3, under the scenario of deterministic sampling without guidance, we rival the generation quality of prior art diffusion models, EDM2. Of note is that the on-par quality is acquired with merely half of the computational load and parameter count. In a horizontal comparison to the GAN family with deterministic sampling, Inception-v3 based FID measurement is blamed for unfairly favoring GANs rather than diffusion models (Stein et al., 2023). Therefore, previous diffusion models have to exchange more than double parameters for lower FID-50K than the best-in-class StyleGAN-XL. Still with better FID, EDM2+, among the diffusion models, is the first to preserve the same magnitude of model size as StyleGAN-XL.

*Comparison to stochastic sampling.* Although stochastic sampling is still at the forefront of cutting-edge diffusion models, it suffers from laborious parameter tuning and a cumbersome sampling trajectory. To outperform EDM1 using stochastic sampling, EDM2-L using deterministic sampling spends nearly triple FLOPs on a single model evaluation, partially canceling out the benefit of fewer sampling steps, while our EDM2+-L could limit the single model FLOPs to the same level as EDM1. In lieu of stochastic sampling, some concurrent works push the performance frontier of deterministic sampling with advanced Classifier-Free Guidance (CFG) (Ho & Salimans, 2021) techniques, such as guidance interval (Kynkäänniemi et al., 2024) and autoguidance (Karras et al., 2024a). Now that this work on neural architecture design is orthogonal to them, our generation performance is

Table 4: Runtime and memory profiling. CPU latency is timed with a batch size of 1 while the GPU throughput with a batch size of 32. The GPU throughput is measured in Frames Per Second (FPS).

| Architecture | CPU Latency (s)↓ | GPU Throughput (img/s)↑ | GPU Memory (MB)↓ |
|---|---|---|---|
| EDM2 | 2.069 | 642 | 1273 |
| EDM2+ | 0.998 *(−52%)* | 836 *(+30%)* | 1114 *(−13%)* |

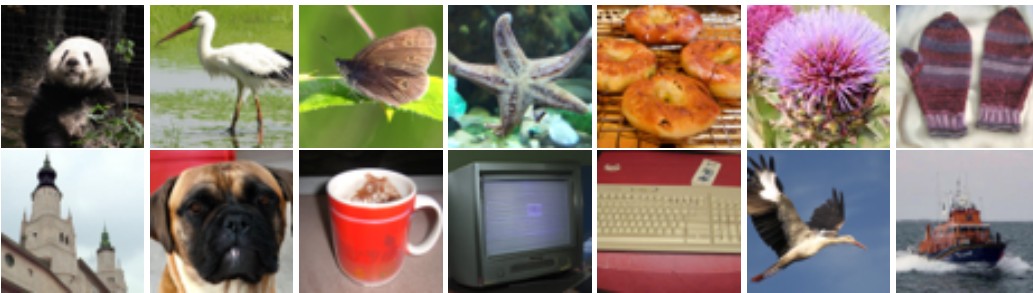

Figure 4: Uncurated images at $64 \times 64$ resolution from EDM2+-XL without guidance.

ready to be further improved using these complementary tricks. As a consequence, the combination of EDM2+ and autoguidance secures an FID of 1.00, reaching the state-of-the-art performance. With evidently fewer sampling steps and total compute, we are able to beat the record FID of 1.23 achieved by stochastic sampling of RIN.

*Runtime analysis.* Figure 1 and Table 3 mainly quantifies the model cost using FLOPs. Nevertheless, it is more practical to inspect the wall clock runtime (Ma et al., 2018). The model inference is profiled on an NVIDIA A100 GPU after warmup, with `torch.compile` and TensorFloat32 (TF32) tensor cores enabled, as well as on an Intel Xeon Platinum 8468V CPU. An apple-to-apple comparison to EDM2 on runtime and memory is collected in Table 4. The hardware execution speed is of great interest, that is improved by 52% on CPU and 30% on GPU device with our EDM2+. As a byproduct, the GPU memory volume during inference time is also shrunk by 13%.

### 4.5 QUALITATIVE RESULTS

We display uncurated class-conditional generation samples in Figure 4. These images are generated with our EDM2+-XL model without guidance. At a glimpse of various samples, the illustrated results embrace a variety of classes represented in the ImageNet dataset, demonstrating great diversity. Looking into each individual sample, though at a low resolution, these synthesized images maintain high fidelity in comparison to real-world photographs.

## 5 CONCLUSION AND LIMITATION

This work invests effort into the rapidly evolving arena of diffusion model architectures, via undertaking a systematic exploration and unraveling practical guidelines for efficient network design. The valuable discoveries, pinpointing the significance of layer placement and module interconnection, are leveraged to deliver a model family named EDM2+. Our presented EDM2+ architecture achieves pronounced efficiency gains against the EDM2 counterpart and redefines the state-of-the-art performance of generative modeling.

Despite promising, the practical runtime is expected to be further optimized for real-time deployment. Probing more fine-grained architecture design options, such as the preference discrepancies between the network encoder and decoder, or even the per-block design regime, is intended as the next step in our research agenda. Neural Architecture Search (NAS) (Zoph & Le, 2017; Zoph et al., 2018) is a plausible avenue to reach this goal. Marrying our EDM2+ architecture to the latent-space diffusion for high-resolution image synthesis is also left as our future work.

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
