# OpenReview forum: "EDM2+: Exploring Efficient Diffusion Model Architectures for Visual Generation"
_ICLR.cc/2025/Conference — Submitted to ICLR 2025_

### Official Review · Reviewer_immE · 2024-10-29

**Soundness:** 2
**Presentation:** 2
**Contribution:** 2
**Rating:** 5
**Confidence:** 4

**Summary:**

In order to solve the problem that the existing diffusion model de-noising network frame parameters are too large, this paper explores the design of the efficient diffusion model EDM2 based on UNET, and explains the role of each component through the ablation experiment of several components. The EDM2++ architecture is redesigned to reduce the computational complexity of EDM2.

**Strengths:**

1. The paper is well-written and easy to understand.
2. A large number of ablation experiments are done to verify the role of each component of the EDM architecture.
3. The experiment of this paper is very sufficient, which fully demonstrates the high efficiency of EDM2+.

**Weaknesses:**

1.There is a lack of adequate theoretical understanding of these components only through ablation experiments. More in-depth theoretical analysis is needed to understand the role of denoising architectures in capturing probability distributions in diffusion models. For example, try to analyze from a geometric point of view. [1]
2.Existing generation models have been extended to a wider range of fields, such as stream matching. The diffusion model can be seen as an example of this. This model framework does not seem to show superior results in other generative paradigms.
[1] GENERALIZATION IN DIFFUSION MODELS ARISES FROM GEOMETRY-ADAPTIVE HARMONIC REPRESENTATIONS .

**Questions:**

1. Does this denoising network model architecture perform well in other generative paradigms?
2. This denoising architecture is very fast. In contrast, many works studying scaling law in diffusion models require large consumption. So what are the limitations of your model compared to these models?

---

> ### Author Response · Authors · 2024-11-24
>
> Thanks for your valuable comments. We clarify what you are concerned about and sincerely hope our detailed feedback is as you expected.
>
> 1. **Generalization to other generative paradigms**
>
> Certainly, the answer is affirmative. In the original manuscript, we follow the popular EDM2 framework to perform experiments in the context of standard diffusion models. In fact, the U-Net style model architecture is widespread in a variety of generative modeling paradigms, including flow matching [1], score-based models [2], autoregressive models [3], and so on.
>
> First, we apply our architectural innovations to the U-Net of a score-based model, following the elegant implementation of `cifar10‑32x32‑cond‑ve` in [EDM](https://github.com/NVlabs/edm). Using ~50% FLOPs, we achieve the same FID of 1.79. Second, we apply our methodology to the U-Net of Pixel CNN++ on CIFAR-10 based on our reimplementation, and get a Bits Per Dimension (BPD) of 2.95 (similar to the published 2.92 in [3]). Third, since flow matching could be interpreted as a simple reweighting of the diffusion loss function [4], it would be straightforward for our network architecture to work with it and obtain a similar FID of 1.60 on ImageNet $64 \times 64$. In summary, our network architecture design is compatible with these different generative modeling paradigms, instead of confined to diffusion models.
>
> 2. **Comparison to large-scale models, *w.r.t* scaling law**
>
> On the one hand, we have already verified our method on large-scale models, *e.g.*, EDM2-XL with over 1.1B parameters in Table 3. The benefit is that our EDM2+-XL is also able to reduce such a large-scale model's parameters and computational complexity to half. This preliminary result demonstrates the potential of our method to reduce the parameters and FLOPs of even larger models. Regarding the scaling law curve, our more efficient model EDM2+ corresponds to a steeper slope than EDM2, *i.e.*, with the same training budget, a lower FID is reached. This rule is also utilized to distinguish enhanced network architecture design in a very recent work [5].
>
> On the other hand, the current limitation is that the performance on ImageNet might saturate even when further scaling up. Thus, the scale of this academic dataset and our currently available computing resources limit the exploration of further scaling. Our research-driven models in this work could still have a gap in visual quality compared to commercial ones, such as Stable Diffusion and Midjourney. In the future, we expect to observe our neural architecture design principles to be applied to a combination of larger production-level models (*e.g.* Stable Diffusion) and massive image-text pair datasets (*e.g.* LAION-5B) in a more diverse and real-world scenario. Unfortunately, we do not have sufficient computing resources and data to conduct this kind of exploration temporarily.
>
> [1] Yaron Lipman, Ricky T. Q. Chen, Heli Ben-Hamu, Maximilian Nickel, Matthew Le. Flow Matching for Generative Modeling. In ICLR, 2023.
>
> [2] Yang Song, Jascha Sohl-Dickstein, Diederik P Kingma, Abhishek Kumar, Stefano Ermon, Ben Poole. Score-Based Generative Modeling through Stochastic Differential Equations. In ICLR, 2021.
>
> [3] Tim Salimans, Andrej Karpathy, Xi Chen, and Diederik P. Kingma. PixelCNN++: Improving the pixelCNN with discretized logistic mixture likelihood and other modifications. In ICLR, 2017.
>
> [4] Diederik Kingma and Ruiqi Gao. Understanding diffusion objectives as the ELBO with simple data augmentation. In NeurIPS, 2023.
>
> [5] Zhengyang Liang, Hao He, Ceyuan Yang, Bo Dai. Scaling Laws For Diffusion Transformers. arXiv:2410.08184, 2024

---

> ### Author Response · Authors · 2024-11-24
>
> 3. **Connection to theoretical analysis**
>
> As recognized by all the reviewers and emphasized in the original manuscript, our primary aim is to advance the fundamental research of building a state-of-the-art efficient neural architecture for diffusion models, which is orthogonal to seminal works towards certain theoretical contributions.
>
> For example, the recommended work [1] reaches the following important and general conclusions:
>
> * DNN denoisers transit from memorization to generalization along with the increasing data size.
> * Given adequate training data, DNN denoisers show strong generalization due to the strong inductive bias.
> * DNN denoisers yield the inductive bias that favors shrinkage in a GAHB.
>
> These general conclusions also apply to our new network architecture and we are happy to include this paper as our related work in the manuscript. Nevertheless, these theoretical conclusions are derived with tiny-scale neural networks (such as U-Net with only 7.6M parameters) and small datasets. Differently, our target is establishing an efficient neural architecture that excels at the ImageNet-level benchmark, which is probably the most challenging image generation benchmark at the academic scale. In addition, our models with hundreds of parameters would be more promising to be put into practical usage, with significantly larger network capacity and SoTA quality on ImageNet.
>
> In this work, we comprehensively explore the network design space of efficient U-Net based diffusion models. Concretely, we systematically study fundamental design choices in terms of ① layer placement inside a denoising network block and ② module interconnection between the denoising network and the embedding network. The specific exploration process and the corresponding findings are elucidated in Section 3. Our empirical insights obtained from these exploration lead to a redesigned architecture, EDM2+, that is $2\times$ more efficient than the prior art EDM2 with on-par or even better generation quality on the challenging ImageNet benchmark. It would be good if the reviewer could elaborate on the exact theoretical analysis required in the scope of this neural network architecture design.
>
> [1] Zahra Kadkhodaie, Florentin Guth, Eero P Simoncelli, Stéphane Mallat. Generalization in diffusion models arises from geometry-adaptive harmonic representations. In ICLR, 2024.

---

> ### Comment · Reviewer_immE · 2024-11-29
>
> Thanks for the response. Even though the authors discuss differences from previous work on generative paradigms and large-scale models, the novelty of this paper is still my major concern. So I decided to keep my rating.

---

> ### Author Response · Authors · 2024-11-30
>
> We sincerely appreciate your continued interest in our work, but we have to reiterate the important content in our responses.
>
> The reviewer stated that "the authors discuss differences from previous work on generative paradigms and large-scale models", which is self-contradictory to your original questions and is very likely a misconception in our view.
>
> First, in response to your Question 1 "Does this denoising network model architecture perform well in other generative paradigms?", we are discussing the **generalization** of our network architecture to other generative paradigms, instead of the **differences**.
>
> Second, in response to your Question 2 "So what are the limitations of your model compared to these models (studying scaling law in diffusion models)?", we are explaining that actually we **have already considered the trend of scaling law** of our proposed model in Table 3 and Figure 1, while the academic-scale ImageNet dataset may constitute a bottleneck of further scaling-up. We are **not discussing the differences** from large-scale models. Instead, our EDM2+ is applicable to large-scale models. Specifically, our EDM2+ still reduces the training and inference cost of 1.1B large-scale EDM2.
>
> We hope the above clarification could further help address the concern of the reviewer. Please let us know if you have other questions and we would be very happy to help answer.

---

### Official Review · Reviewer_4WHB · 2024-10-30

**Soundness:** 4
**Presentation:** 4
**Contribution:** 3
**Rating:** 6
**Confidence:** 4

**Summary:**

This paper systematically studies the architecture design space of the EDM2 model. By substituting standard convolution with depthwise and pointwise convolution and decreasing the condition embedding dimension, the proposed EDM2+ model reduces the computational complexity by 2× (measured by the number of parameters and FLOPs) while maintaining the generation quality (measured by FID).

**Strengths:**

1. The writing of the paper is clear and easy to follow overall.
2. The paper conducts a comprehensive ablation study to evaluate the impact of various design choices on the network architecture. After systematical study and careful design, the final EDM2+ excels on the ImageNet benchmark.

**Weaknesses:**

1. While the reduction in FLOPs and CPU latency along with the increased GPU throughput is remarkable, the depthwise and pointwise convolution operations may not contribute significantly to GPU latency improvements. Including additional results on GPU latency would enhance the comprehensiveness of the experimental findings.
2. The architecture design presented in this paper aligns closely with established efficient network designs, such as the MobileNet and EfficientNet series, which somewhat constrains its novelty. Nonetheless, the exploration within the image generation domain is a noteworthy contribution.

**Questions:**

I do not have other questions.

---

> ### Author Response · Authors · 2024-11-23
>
> We sincerely appreciate your positive feedback. Below, we have tried to answer your questions and concerns.
>
> 1. **GPU runtime**
>
> GPU throughput is a very widely adopted metric for speed comparison, across image recognition models [1, 2] and generation models [3, 4]. First, for image synthesis, generating images in a batch is also a common practice, so we consider it of practical value to regard GPU throughput as the primary metric in the original manuscript. Second, it is also stated in [4] that "Under the circumstances that GPU is fully utilized, the measurement is more meaningful." In this context, batched inference makes more sense since the GPU latency measured with a batch size of 1 does not fully exploit the GPU utility and ends up returning a non-linear scaling behavior. Since we put our emphasis on efficient architectures that are friendly to GPU deployment, we measure GPU throughput with a batch size of 32.
>
> As a supplement, we also measure the GPU latency on an NVIDIA A100 GPU and consider it as the secondary metric. The GPU latency is 0.0217s for EDM2 with 700M GPU memory consumed, while it is 0.0226s for EDM2+ with 462M memory consumed. When batch size is 1, our architecture achieves a similar performance-runtime trade-off, occupying remarkably less GPU memory.
>
> 2. **Comparison to MobileNet/EfficientNet**
>
> In a word, they are superficially similar but essentially different. Our final design is a better architecture choice based on our extensive experiments.
>
> As acknowledged by all the reviewers and highlighted in the original manuscript, our core contribution is comprehensively investigating the design space of efficient U-Net architecture for diffusion models, specifically the role of each component in a network block, and further enhancing the existing design significantly. Note that we do not simply substitute standard convolution with depthwise and pointwise convolution. Instead, we thoroughly explore the design space from the perspective of layer placement and module interconnection, as mentioned in Line 18-19 (Abstract), Line 91 (Introduction), and Section 3.2-3.4.
>
> On the one hand, as stated in Line 245-247, it is very common that depthwise and pointwise convolution exist in any efficient network architectures. As of the year 2024, we would like to take them as basic primitives for efficient networks, just like vanilla convolution for plain neural networks. Without exception, the classical MobileNet and EfficientNet series also extensively use depthwise and pointwise convolution in their architectures, so this is why they superficially look similar to ours.
>
> On the other hand, the major distinction is that we are pushing the performance-efficiency frontier and designing novel diffusion model architecture with these operators. Please kindly notice that, given the depthwise and pointwise convolution operators, there exist tons of configurations that can give rise to new architectures. Both MobileNet and EfficientNet build upon a specific configuration, MBConv blocks. It is noteworthy that CONFIG E, an intermediate variant during our exploration journey, coincides with MBConv, as stated in Line 306-308. However, its performance-efficiency trade-off is still suboptimal in our evaluation (please see Table 1), compared to our final status in CONFIG G. This comparison demonstrates that the specific design for representative recognition models is *not* naively transferable to generative models. Thus, we make more efforts to delve deeper into the architecture design of diffusion models. Moreover, we also explore the design of embedding network, that does not appear in recognition models like MobileNet or EfficientNet but contributes a lot to our architecture. As you have commented, "The paper conducts a comprehensive ablation study to evaluate the impact of various design choices on the network architecture."
>
> Last but not least, as you have mentioned, "the exploration within the image generation domain is a noteworthy contribution". Our efforts fill in the significant gap, providing guidelines for fundamental architecture design of diffusion models, including but not limited to how to better employ depthwise and pointwise convolutions for improved efficacy and efficiency. We believe it will interest a broad audience in the generative modeling community.
>
> [1] Ze Liu, Yutong Lin, Yue Cao, Han Hu, Yixuan Wei, Zheng Zhang, Stephen Lin, Baining Guo. Swin Transformer: Hierarchical Vision Transformer Using Shifted Windows. In ICCV, 2021.
>
> [2] Zhuang Liu, Hanzi Mao, Chao-Yuan Wu, Christoph Feichtenhofer, Trevor Darrell, Saining Xie. A ConvNet for the 2020s. In CVPR, 2022.
>
> [3] Xinyin Ma, Gongfan Fang, Xinchao Wang. DeepCache: Accelerating Diffusion Models for Free. In CVPR, 2024.
>
> [4] Anil Kag, Huseyin Coskun, Jierun Chen, Junli Cao, Willi Menapace, Aliaksandr Siarohin, Sergey Tulyakov, Jian Ren. AsCAN: Asymmetric Convolution-Attention Networks for Efficient Recognition and Generation. In NeurIPS, 2024.

---

> > ### Comment · Reviewer_4WHB · 2024-11-25
> >
> > Thank you for the detailed response. I appreciate that the authors have addressed my questions. I will maintain my score of 6.

---

### Official Review · Reviewer_pkLA · 2024-11-03

**Soundness:** 3
**Presentation:** 3
**Contribution:** 2
**Rating:** 6
**Confidence:** 2

**Summary:**

This paper explores the design landscape of efficient U-Net based diffusion models through comprehensive experiments based on EDM2. The findings can be summarized into two main points: first, decomposing the spatial/channel mixing operations and shifting the computation focus from spatial to channel dimension strikes a better balance; second, the key to both efficacy and efficiency is modulating the condition embedding into the denoising network’s bottleneck layers.

**Strengths:**

1. The efficient diffusion model architectures reduce the computing resources required for experiments, which is beneficial for further research in related fields.

2. The results on ImageNet are impressive, as they maintain generation quality while reducing computational complexity by about half.

3. The experiments are well-designed and conducted, incorporating the conventional wisdom of previous classic works.

**Weaknesses:**

1. How do different architectures affect the training time?
2. Curious about whether the proposed architectures could be generalized to accelerate the training of consistency models [1, 2]. ECT [2] is also based on EDM and EDM2, though this may still necessitate comprehensive implementation and is not a primary part of my decision assessment.

References:

[1] Song, Yang, and Prafulla Dhariwal. "Improved techniques for training consistency models." *arXiv preprint arXiv:2310.14189* (2023).

[2] Geng, Zhengyang, et al. "Consistency Models Made Easy." *arXiv preprint arXiv:2406.14548* (2024).

**Questions:**

See above.

---

> ### Author Response · Authors · 2024-11-23
>
> We appreciate the reviewer for your constructive comments and provide additional discussions in the following to complement the original manuscript.
>
> 1. **Training time**
>
> The wall clock training time of EDM2+ is <90% of EDM2. It has been reported that the GPU throughput shows a speedup of 30% in Table 4. During training, considering the same data loading pipeline, communication cost across GPU devices, and other common factors between different architectures, the overall training speedup ratio is reasonable. In addition, one would be usually concerned more about the inference time during deployment in comparison to the training time.
>
> Theoretically speaking, the training time mainly includes network forward & backward time. The training compute can be roughly estimated as $3\times$ model FLOPs, according to [1]. With theoretically less FLOPs, EDM2+ also saves training budget in practice. As your comments in Strength 1, "The efficient diffusion model architectures reduce the computing resources required for experiments, which is beneficial for further research in related fields."
>
> 2. **Application to consistency models**
>
> The answer is affirmative. Interestingly, ECT is a contemporary work to us, being submitted to the same ICLR 2025 venue. As you mentioned, we have not yet conducted comprehensive experiments on top of ECT. Based on their public code, we implement ECT based on EDM2+-S, and achieve an FID of 3.20 with a 2-step generation, very similar to the ECM-S' performance in Table 1 of [2]. Again, the main advantage compared to ECM-S (EDM2-S with ECT) is the improved efficiency. The experiment result speaks for the generalization ability of our EDM2+ architecture, in the context of few-step diffusion distillation setting.
>
> [1] William Peebles, Saining Xie. Scalable Diffusion Models with Transformers. In ICCV, 2023.
>
> [2] Zhengyang Geng, Ashwini Pokle, William Luo, Justin Lin, J. Zico Kolter. Consistency Models Made Easy. arXiv:2406.14548, 2024.

---

### Official Review · Reviewer_NMQM · 2024-11-05

**Soundness:** 2
**Presentation:** 2
**Contribution:** 2
**Rating:** 3
**Confidence:** 4

**Summary:**

This paper presents EDM2+, an exploration of efficient U-Net based diffusion model architectures that builds upon EDM2. While prior work has extensively studied training and sampling of diffusion models, the underlying network architecture design remains underexplored. The authors investigate the design landscape along two key axes - layer placement and module interconnection - leading to several critical insights for superior efficiency. Through systematic analysis and ablation studies, they develop EDM2+, which reduces the computational complexity of EDM2 by 2× without compromising generation quality.
The authors' main contributions include: (1) conducting comprehensive experiments to identify EDM2's architectural limitations, (2) conceptualizing performance-optimized solutions for both generation quality and efficiency, and (3) developing an architecture that excels other leading diffusion models and GANs on the ImageNet benchmark. When equipped with autoguidance, EDM2+ sets new record FID scores on ImageNet 64×64 using deterministic sampling, offering a new standard in the field of generative modeling.

**Strengths:**

- Clear demonstration of computational savings (2× reduction in FLOPs)
- Practical improvements in both CPU and GPU runtime performance
- Systematic exploration of depthwise separable convolutions in diffusion context
- Practical consideration of both training and inference efficiency

**Weaknesses:**

Critical gaps in theoretical understanding:
- No rigorous analysis of why channel mixing becomes more important under computational constraints
- Missing theoretical justification for embedding bottleneck design
- Limited connection to existing theory on efficient architecture design
- Insufficient analysis of how these modifications interact with diffusion dynamics

Incomplete Experimental Analysis:
- No parameter interpolation studies showing trends and sweet spots for architectural design.
- Absence of scaling law analysis across different model sizes
- Limited investigation of failure modes and edge cases
- Missing analysis of how modifications affect different stages of the diffusion process

Others:
- Most architectural choices seem derived from existing efficient CNN literature without sufficient adaptation to diffusion-specific challenges
- Minimal discussion of why certain modifications work better than others
- Few transferable insights compared to previous works like EDM2
- Limited exploration of trade-off spaces

**Questions:**

The paper presents results for EDM2+-S, L, and XL variants, but notably omits the medium-sized (M) model experiments that were present in the original EDM2 paper. Could you explain the reasoning behind this omission?

---

> ### Author Response · Authors · 2024-11-24
>
> Thanks for your recognition of our network architecture's performance and your detailed feedback. We would like to clarify certain details by answering your questions one by one.
>
> **Medium-sized model**
>
> First, faithfully following the training recipe of EDM2, the medium-sized model consumes the most training cost across all model sizes (7.8 days according to Table 6 in the appendix of [1]). Due to the limited computational resources before submission, we mainly conduct ablation experiments on the Small-sized model (S). In order to verify the scaling-up characteristics of our model and conduct a system-level comparison, we additionally run EDM2+-L and EDM2+-XL, as shown in Table 3 in the original manuscript. As shown in Figure 1, with access to the results of EDM2+-S and EDM2+-L/XL, we are able to plot the two endpoints of the performance-efficiency curve.
>
> Second, after submission, we leverage the available computing resources to run EDM2+-M. It achieves an FID of 1.43 with 273M parameters and 89GFLOPs, totally consistent with the conclusion reported in our original manuscript.
>
> [1] Tero Karras, Miika Aittala, Jaakko Lehtinen, Janne Hellsten, Timo Aila, and Samuli Laine. Analyzing
> and improving the training dynamics of diffusion models. In CVPR, 2024.

---

> ### Author Response · Authors · 2024-11-24
>
> **Critical gaps in theoretical understanding**
>
> As recognized by all the reviewers and emphasized in the original manuscript, our primary aim is to advance the fundamental research of building a state-of-the-art efficient neural architecture for diffusion models. It is orthogonal to seminal works towards certain theoretical contributions.
>
> For example, the recommended work [1] by Reviewer immE reaches several important and general conclusions. These conclusions also apply to our new network architecture and we are happy to include this paper as our related work in the manuscript. Nevertheless, most of these theoretical conclusions are derived from tiny-scale neural networks (such as U-Net with only 7.6M parameters) and toy datasets. Differently, our target is establishing an efficient neural architecture that excels at the ImageNet-level benchmark, which is probably the most challenging image generation benchmark at the academic scale. In addition, our models with hundreds of parameters would be more promising to be put into practical usage, with significantly larger network capacity and SoTA quality on ImageNet.
>
> In this work, we comprehensively explore the network design space of efficient U-Net based diffusion models. Concretely, we systematically study fundamental design choices in terms of ① layer placement inside a denoising network block and ② module interconnection between the denoising network and the embedding network. The specific exploration process and the corresponding findings are elucidated in Section 3. Our empirical insights obtained from these explorations lead to a redesigned architecture, EDM2+, that is $2\times$ more efficient than the prior art EDM2 with on-par or even better generation quality on the challenging ImageNet benchmark. To be specific,
>
> * Our statement is that channel mixing outweighs spatial mixing in the considered image synthesis domain. There exists a more detailed description and explanation in Line 246-252. This statement is supported by a series of comparisons in CONFIG A-E. Overall, among the experiments of CONFIG A-E, reducing the usage of spatial mixing while shifting the computation focus to the channel dimension usually strikes a better performance-efficiency balance.
>
> * We have analyzed the embedding bottleneck design in Section 3.4, particularly in Line 364-372. Following previous works, the embedding network is gradually miniaturized with on-par or even stronger performance. Integrating the condition information into a narrow feature map would be more effective, yielding more targeted and predictable control of the entire information flow. The comparison between CONFIG F-G and CONFIG F\*-G\* in Table 2 clearly demonstrates the efficacy and efficiency brought by the simple yet effective embedding network design.
>
> * Our EDM2+ indeed has a connection to the existing theory of efficient network architecture design. As in Reviewer pkLA's Strength, "The experiments are well-designed and conducted, incorporating the conventional wisdom of previous classic works.". We present some relationships dispersedly in Line 245-247, Line 262-266, and Line 269-291. A more comprehensive compilation of the similarities and differences is deferred to our response later, in terms of "Others".
>
> * As you commented in the Strength part, we conduct a "Systematic exploration of depthwise separable convolutions in diffusion context". Our developed network architecture is encompassed in the EDM preconditioning framework, as mentioned in Line 138-140, where most of the diffusion dynamics follow EDM1 or EDM2, such as the training schedule, sampling schedule, and prediction target parameterization. Throughout our paper, a great synergy is achieved between each of our modifications to the network architecture and the underlying diffusion dynamics.
>
> [1] Zahra Kadkhodaie, Florentin Guth, Eero P Simoncelli, Stéphane Mallat. Generalization in diffusion models arises from geometry-adaptive harmonic representations. In ICLR, 2024.

---

> ### Author Response · Authors · 2024-11-24
>
> **Incomplete Experimental Analysis**
>
> Although all the other reviewers reach a consensus that the experiments are abundant (Reviewer immE: "The experiment of this paper is very sufficient"; Reviewer 4WHB: "The paper conducts a comprehensive ablation study"; Reviewer pkLA: "The experiments are well-designed and conducted"), we would like to further clarify certain details in the following.
>
> * Actually, we have already performed experiments on EDM2+ model variants with different parameter counts, as shown in Table 3. Specifically, for the ImageNet dataset, the sweet spot is EDM2+-L, observed from Figure 1. Since the dataset size of ImageNet is limited, the performance saturates when further scaling up the model size. This observation also applies to the original EDM2.
>
> * The scaling property is intuitively demonstrated in Figure 1. As mentioned above, the FID decreases along with the increasing parameters and computational complexity at first. After a certain point, the FID plateaus. It does not mean that the scaling law is invalid here. Instead, the dataset size becomes the bottleneck.
>
> * Even the current state-of-the-art image generation models can still produce images with notable artifacts, which has been widely noticed, especially when trained on the controlled, academic data such as ImageNet. To name a few, failure modes or edge cases include
>
>     * [unnatural shape](https://anonymous.4open.science/r/edm2p-9FD1/shape.png): too long vehicle, irregular lane
>
>     * [composition of multiple object parts](https://anonymous.4open.science/r/edm2p-9FD1/composition.png): two dog heads
>
>     * [irrational facial features of a human face](https://anonymous.4open.science/r/edm2p-9FD1/face.png)
>
>     * [unrecognizable texture or patterns](https://anonymous.4open.science/r/edm2p-9FD1/texture.png)
>
> * We find the stage-wise diffusion losses are similar between EDM2 and EDM2+, while the major advantage of EDM2+ is its efficiency. EDM2+ achieves the same magnitude of loss with half of the parameters and computational complexity of EDM2.

---

> ### Author Response · Authors · 2024-11-24
>
> **Others**
>
> * In a word, the architectural choices are superficially similar to existing efficient CNN literature but are essentially different at a close look. Moreover, our final design is a better architecture choice based on our extensive experiments, specifically suitable for diffusion models.
>
>   As acknowledged by all the reviewers and highlighted in the original manuscript, our core contribution is comprehensively investigating the design space of efficient U-Net architecture for diffusion models, specifically the role of each component in a network block, and further enhancing the existing design significantly. Note that we do not simply substitute standard convolution with depthwise and pointwise convolution. Instead, we thoroughly explore the design space from the perspective of layer placement and module interconnection, as mentioned in Line 18-19 (Abstract), Line 91 (Introduction), and Section 3.2-3.4.
>
>   On the one hand, as stated in Line 245-247, it is very common that depthwise and pointwise convolution exist in any efficient network architectures. As of the year 2024, we would like to take them as basic primitives for efficient networks, just like vanilla convolution for plain neural networks. Without exception, many existing efficient CNN also extensively use depthwise and pointwise convolution in their architectures, so this is why they superficially look similar to ours.
>
>   On the other hand, the major distinction is that we are pushing the performance-efficiency frontier and designing novel diffusion model architecture with these operators. Please kindly notice that, given the depthwise and pointwise convolution operators, there exist tons of configurations that can give rise to new architectures. For example, both MobileNet and EfficientNet build upon a specific configuration, MBConv blocks. It is noteworthy that CONFIG E, an intermediate variant during our exploration journey, coincides with MBConv, as stated in Line 306-308. However, its performance-efficiency trade-off is still suboptimal in our evaluation (please see Table 1), compared to our final status in CONFIG G. This comparison demonstrates that the specific design for representative recognition models is *not* naively transferable to generative diffusion models. Thus, we make more efforts to delve deeper into the architecture design of diffusion models. Moreover, we also explore the design of embedding network, that does not appear in recognition models but contributes a lot to our diffusion model architecture. These are all diffusion-specific challenges tackled in our work.
>
>   Last but not least, our efforts fill in the significant gap, providing guidelines for fundamental architecture design of diffusion models, including but not limited to how to better employ depthwise and pointwise convolutions for improved efficacy and efficiency. As Reviewer 4WHB's comments, "the exploration within the image generation domain is a noteworthy contribution". We believe it will interest a broad audience in the generative modeling community.
>
> * It is unclear why the reviewer considers "minimal discussion of certain modifications" in this work. In fact, we perform many comparative experiments in the main body (Section 3). Each revision is enabled after full discussions. As stated in Line 214-215, "the revisions are enabled one by one on top of the baseline EDM2 and we delineate the stepwise quality metric accompanied by the model parameters and computational complexity." Moreover, all the other reviewers have already acknowledged that the ablation experiments are abundant (Reviewer 4WHB: "The paper conducts a comprehensive ablation study"; Reviewer pkLA: "The experiments are well-designed and conducted"; Reviewer immE: "The experiment of this paper is very sufficient").
>
> * Overall, the EDM2+ network architecture is based on U-Net, which is widely used in academia and industry as the diffusion model backbone, as well as in many other low-level vision tasks. These U-Net models are also commonly composed of basic blocks. Conditional generation models typically include an embedding network. Thus, the design philosophy of improved basic block and embedding network in this work is ready to be transferred to other similar models. In addition, please refer to our response to Reviewer immE in terms of "Generalization to other generative paradigms", the network architecture is compatible with a variety of generative modeling paradigms. Thus, the versatility and transferability of our netwok architecture has been proved to some extent.
>
> * On the one hand, regarding to a roughly fixed model capacity level, we are essentially seeking a (near-)optimal trade-off during the whole exploration journey in Section 3. On the other hand, regarding to a range of model capacity levels, we perform a bunch of experiments with a spectrum of model sizes in Section 4. These two categories both belong to the exploration of the trade-off spaces.

---

> > ### Comment · Reviewer_NMQM · 2024-11-27
> >
> > I appreciate the authors' detailed response addressing the concerns raised in my initial review. While the response clarifies several aspects of the work and demonstrates the practical benefits of EDM2+, I would like to share my remaining concerns.
> >
> > First, regarding the theoretical foundations, while the empirical results from CONFIG A-E demonstrate the superiority of channel mixing under computational constraints, the response still lacks a rigorous theoretical explanation for this phenomenon. A deeper analysis connecting these architectural choices to fundamental principles would significantly strengthen the paper's contribution.
> >
> > The authors argue that the architectural modifications, particularly in the arrangement of depthwise/pointwise convolutions, are "superficially similar but essentially different" from existing efficient CNN literature. However, without a clear theoretical framework explaining why these specific arrangements are particularly suited for diffusion models, the differences appear somewhat incremental. Understanding how these architectural choices specifically address diffusion-related challenges would help establish the uniqueness of the approach.
> >
> > Regarding the embedding bottleneck design, while the response suggests it provides "more targeted and predictable control," a more mechanistic explanation would be valuable. A clearer understanding of how this design choice influences the diffusion process could provide important insights for future architecture development.
> >
> > I agree with other reviewers that this work conducts extensive ablation studies, which is essential for empirical research. However, my concern lies with the nature of the analysis - while the quantitative experiments are thorough, there is limited qualitative analysis explaining why each modification proves effective. This deeper understanding would be crucial for advancing the field of efficient diffusion architectures.
> >
> > I believe empirical studies should be accompanied by theoretical insights that can guide future research. It is in this spirit that I maintain my previous assessment, hoping to encourage work that combines strong practical results with deeper theoretical understanding. Therefore, I maintain my original rating.

---

> > > ### Author Response · Authors · 2024-11-30
> > >
> > > Thanks for your recognition of our previous responses and agreement with our contributions to empirical research.
> > >
> > > First, as you mentioned, "empirical studies should be accompanied by theoretical insights that can guide future research". We have indeed provided preliminary network design insights in our work, and leverage these insights to guide the final architecture design of EDM2+, which is one of our contributions highlighted in Line 19-21 and elaborated in Section 3. The specific explanations are reiterated in the first three bullets in our previous response https://openreview.net/forum?id=T1MTmAlF7x&noteId=lwEXUZHBlu.
> > >
> > > Second,  in terms of ① layer placement inside a denoising network block and ② module interconnection between the denoising network and the embedding network, it would be good if the reviewer could elaborate on the exact theoretical points required in the scope of this neural network architecture design. As a strong reference, the well-known work EDM2 [1] also almost verifies the improvement of architecture design empirically. It still attracts a lot of attention within the generative modeling community.
> > >
> > > Third, per the ICLR 2025 Reviewer Guide https://iclr.cc/Conferences/2025/ReviewerGuide, please kindly consider "Objective of the work: What is the goal of the paper? Is it to better address a known application or problem, draw attention to a new application or problem, or to introduce and/or explain a new theoretical finding? A combination of these? Different objectives will require different considerations as to potential value and impact." From our understanding, it is NOT a strict requirement that one work putting more emphasis on empirical verification should also simultaneously provide an in-depth theoretical analysis. Therefore, we respectfully beg to differ with the reviewer that "It is in this spirit that I maintain my previous assessment, hoping to encourage work that combines strong practical results with deeper theoretical understanding."
> > >
> > > We hope  the above clarification could further help address the concern of the reviewer. Please let us know if you have other questions and we would be very happy to help answer.
> > >
> > > [1] Tero Karras, Miika Aittala, Jaakko Lehtinen, Janne Hellsten, Timo Aila, and Samuli Laine. Analyzing
> > > and improving the training dynamics of diffusion models. In CVPR, 2024.

---

### Meta-Review · Area_Chair_w7Rn · 2024-12-19

**Metareview:**

This paper examines effective architectures for diffusion methods. The evaluation scores remain inconsistent after a constructive discussion between authors and reviewers. The major remaining issues are experiments and novelty.  Consequently, I have personally reviewed the paper. Overall, I believe it falls short of the quality needed for publication. The primary reasons are outlined below:

- **Experiments**: This research mainly evaluates their approach using the Imagenet dataset with a fixed and limited resolution. It is still unclear if the proposed method can sustain a performance/efficiency balance in more complex scenarios. I have considered the review guidelines to evaluate a paper based on its objective.   But, for an empirical paper, I think a thorough and extensive empirical study is necessary.
- **Contributions**: The innovation is somewhat limited. As the authors pointed out in the paper, the proposed method mainly integrates established practices.  Again, I have considered the guidelines to rule out the impact of theoretical weakness.

Thus, I have to reject this paper. I encourage the authors to investigate architectures in more complicated scenarios and datasets to demonstrate a generalizable empirical result.

**Additional Comments On Reviewer Discussion:**

`pkLA` and `4WHB` are weakly positive towards this paper. I agree that the proposed method has improved performance on the reported dataset.

But my decision is based on the following issues:

- `NMQM` believes this paper requires theoretical justifications. It might be too harsh to require an application paper to include a solid theory. However, the paper should at least include sufficient intuitive explanations of their contribution, especially when the authors claimed that their work mainly integrates established practices.

- ` immE` believes the experiment results should be extended to other datasets and tasks. I agree with this since the original paper only tests its work merely on the Imagenet dataset with a fixed and limited resolution. Since the author has positioned their work as an application study, thorough empirical analysis must be conducted across different settings and datasets. Unfortunately, the authors didn't provide further experiments during the rebuttal.

---

### Decision · Program_Chairs · 2025-01-22

Reject